# Major adverse cardiovascular events and hyperuricemia during tuberculosis treatment

Hong-Joon Shin[1,2], Joon-Young Yoon[1], Young-Ok Na[1], Jae-Kyeong Lee[1], Bo Gun Kho[1], Tae-Ok Kim[1,2], Yu-Il Kim[1,2], Sung-Chul Lim[1,2], Sae-Hee Jeong[1], Yong-Soo Kwon[1,2] *

1 Department of Internal Medicine, Chonnam National University Hospital, Gwangju, Republic of Korea,
2 Chonnam National University Medical School, Gwangju, Republic of Korea

* yskwon@jnu.ac.kr

**Data Availability Statement:** All relevant data are within the paper and its Supporting information files.

## Abstract

### Background

Hyperuricemia is common during tuberculosis (TB) treatment, especially in association with pyrazinamide (PZA). This study investigated the relationship between major adverse cardiovascular events (MACEs) and hyperuricemia during TB treatment.

### Methods

We conducted a single-center retrospective cohort study. From January 2010 through June 2017, we assessed all consecutive TB patients at Chonnam National University Hospital in South Korea. Hyperuricemia was defined as serum uric acid levels exceeding 7.0 mg/dL (men) and 6.0 mg/dL (women).

### Results

Of the 1,143 patients included, PZA was administered to 1,081 (94.6%), and hyperuricemia was detected in 941 (82.3%). Eight patients experienced MACEs. Multivariate analysis using logistic regression indicated that prior ischemic heart disease was associated with MACE development (OR,14.087; 95% CI,3.304–60.061; P < 0.000), while hyperuricemia was not (OR, 1.505; 95% CI, 0.184–12.299; P = 0.703). For patients without drug-resistant TB, the absence of hyperuricemia was associated with higher mortality (OR, 2.609; 95% CI, 1.066–6.389; P = 0.036), whereas hyperuricemia was associated with less worse outcomes (OR,0.316; 95% CI,0.173–0.576; P < 0.000).

### Conclusions

Although most patients treated with PZA developed hyperuricemia, it was not associated with MACE development. Hyperuricemia during TB treatment was associated with better outcomes, possibly due to consistent adherence to TB treatment.

**Funding:** YSK received a grant (BCRI20013) of Chonnam National University Hospital Biomedical Research Institute. The funders had no role in study design, data collection and analysis, decision to publish, or preparation of the manuscript.

**Competing interests:** The authors have declared that no competing interests exist.

# Introduction

Tuberculosis (TB) is a preventable and curable disease when treatment is administered appropriately. Nonetheless, it remains a significant global health issue. As per the 2022 World Health Organization (WHO) TB report, 6.4 million people were diagnosed with TB, with 1.6 million deaths from TB in 2021 [1]. One of the main hurdles to successful TB treatment is adverse drug reactions (ADRs) resulting from anti-TB medications. The standard treatment for drug-susceptible TB, which includes isoniazid, rifampin, ethambutol (EMB), and pyrazinamide (PZA), has shown good efficacy. However, several ADRs are known to manifest during treatment with this regimen [2]. Among the ADRs, hyperuricemia is mainly caused by PZA. Administering PZA during the initial intensive phase increases serum uric acid levels by decreasing renal uric acid clearance [3, 4]. If conditions like gout or renal lithiasis arise, PZA administration should be discontinued.

Though asymptomatic hyperuricemia does not necessitate separate treatment, it has been consistently associated with cardiovascular disease [5–7]. Many recent studies have suggested that hyperuricemia increases the risk of cardiovascular disease, including hypertension, major adverse cardiac events (MACEs), and metabolic syndrome [8–10]. However, some counterarguments suggest uric acid does not cause cardiovascular disease [11]. Hyperuricemia is caused by an increase in uric acid production due to factors such as increased meat consumption, tumor lysis syndrome, and hemolytic anemia, or a decline in renal excretion owing to renal dysfunction and certain drugs [12–14]. Anti-TB drugs like EMB and PZA, by diminishing renal uric acid clearance, lead to reversible hyperuricemia [3, 4]. This study aimed to evaluate the association between major adverse cardiovascular events (MACEs) and hyperuricemia during TB treatment.

# Methods

## Study sample

This study was a retrospective single-center case-control study. We reviewed the data of all patients aged over 18 years who were registered and treated for pulmonary TB or extrapulmonary TB at Chonnam National University Hospital from January 2010 through June 2017. We accessed the data for research purposes from February 22, 2023. Patients without serum uric acid level results during the first 2 months of TB treatment were excluded. Patients who died, as well as those lost to follow-up or transferred out within 2 months of TB treatment, were also excluded. TB was diagnosed based on *Mycobacterium tuberculosis* identification by culture or polymerase chain reaction from clinical specimens or by clinical, radiological, or histological findings compatible with TB and responses to TB treatment.

In South Korea, the standard treatment for drug-susceptible TB comprises a 6-month self-administered regimen, including a 2-month initial phase of isoniazid, rifampicin, PZA, and EMB, followed by a 4-month continuation phase of isoniazid, rifampicin, and EMB, as recommended by the local guidelines. An alternative 9-month regimen with isoniazid, rifampin, and EMB can be administered. EMB can be stopped if isolated *M. tuberculosis* is susceptible to isoniazid, rifampin, EMB, and PZA. Therefore, most patients in this study received daily therapy consisting of isoniazid (300 mg), rifampin (450 to 600 mg), EMB (0 to 1200 mg), and PZA (1500 mg). However, the clinicians' decision to include PZA in the initial regimen was based on each patient's clinical situation.

Routine laboratory tests, including complete blood count, liver function tests, and renal function tests, were conducted at baseline and each clinical visit according to the local

guidelines. At our institution, serum uric acid levels were included in the follow-up laboratory tests regardless of symptoms (such as arthralgia and arthritis).

## Data collection

We retrospectively reviewed the medical records and laboratory findings of the enrolled patients during the study period. This review included patient age, sex, body mass index (BMI), underlying diseases (hypertension, diabetes mellitus, dyslipidemia, previous ischemic heart disease, atrial fibrillation, chronic kidney disease, chronic obstructive pulmonary disease, and gout), initial treatment regimens, baseline and follow-up laboratory findings (hemoglobin, C-reactive protein [CRP], creatinine, and serum uric acid levels), treatment outcomes, and the occurrence of MACEs or death.

## Definitions

Hyperuricemia was defined as serum uric acid > 7.0 mg/dL for men and > 6.0 mg/dL for women during TB treatment [8, 15, 16]. MACEs were defined as the composite of cardiovascular death, myocardial infarction, angina, stroke, transient ischemic attack, and myocardial revascularization [17].

For the analysis of treatment outcomes, we applied definitions from the WHO recommendations regarding cure, treatment completed, treatment failed, death, and loss to follow-up [18]. Cure and treatment completion were categorized as favorable outcomes, respectively, while treatment failure, death, and loss to follow-up were categorized as unfavorable outcomes.

Anemia was defined according to the WHO guidelines: hemoglobin < 13.0 mg/dL for men and < 12.0 mg/dL for women [19].

## Statistical analysis

Data are expressed as mean (standard deviation), median (interquartile range), or number (percentage). Student's t-tests (for continuous variables) and chi-square tests (for categorical variables) were used to compare the characteristics of patients with and without hyperuricemia. Student's t-tests or Mann-Whitney U tests (for continuous variables), or chi-square tests (for categorical variables) were used to compare the characteristics of patients with and without MACEs. Univariate logistic regression analysis identified factors associated with MACEs, death, and favorable outcomes. Subsequent multivariate logistic regression analyses incorporated variables with P values < 0.25 from the univariate analysis. Statistical analyses were conducted using SPSS Statistics for Windows, version 25.0 (IBM Corp., Armonk, NY, USA). A P value <0.05 was considered statistically significant.

## Ethics statement

The study protocol was approved by the Institutional Review Board (IRB) of Chonnam National University Hospital (approval number: 2023–045). The research adhered to the principles of the Declaration of Helsinki. All methods in this study complied with relevant guidelines and regulations. The IRB of Chonnam National University Hospital waived the requirement for informed consent due to the minimal risk and the retrospective nature of the study, and the patient information was anonymized prior to the analysis.

## Results

A total of 1,674 patients treated for TB were screened during the study period (Fig 1). Of these, 531 were excluded because they were lost to follow-up or transferred out within 2 months of

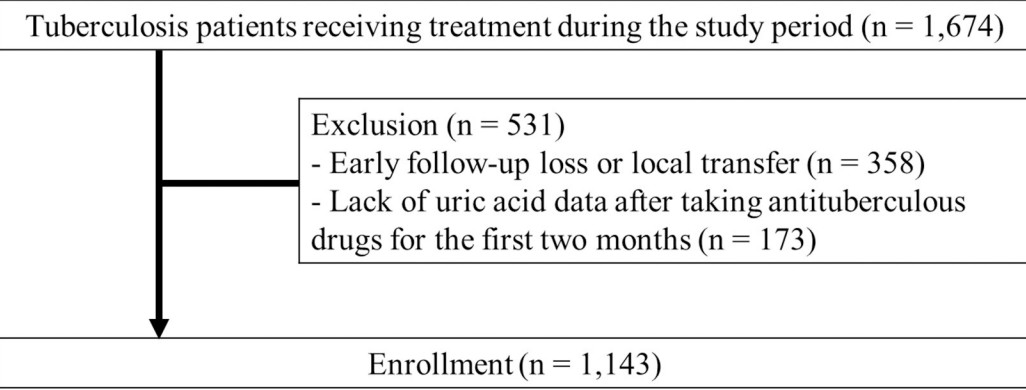

**Fig 1. Patient inclusion and exclusion flow chart.** Out of the 1,674 tuberculosis patients treated at our hospital, 531 were excluded, leading to a final count of 1,143 patients in this study.

TB treatment (n = 358) or because of an absence of serum uric acid data (n = 173). Therefore, 1,143 patients were included in this study.

Of the 1,143 patients included, PZA was administered to 1,081 (94.6%), and hyperuricemia was detected in 941 (82.3%). In patients who used PZA, serum uric acid levels and frequencies of hyperuricemia were significantly higher at 2, 4, and 6 months after initiating TB treatment (Fig 2).

Of the 374 patients with baseline serum uric acid levels, 308 (82.4%) exhibited hyperuricemia at the 2-month mark of their TB treatment. By the end of the treatment, however, the rate of hyperuricemia dropped substantially, approximating the baseline rate (Fig 3).

To determine whether EMB could increase serum uric acid levels and the frequency of hyperuricemia, we evaluated serum uric acid levels and hyperuricemia in patients treated with the combination of isoniazid, rifampin, and EMB or the combination of isoniazid and rifampin, 2 months after starting TB treatment. The levels of serum uric acid and the frequency of hyperuricemia did not differ statistically between the two groups (S1 and S2 Figs).

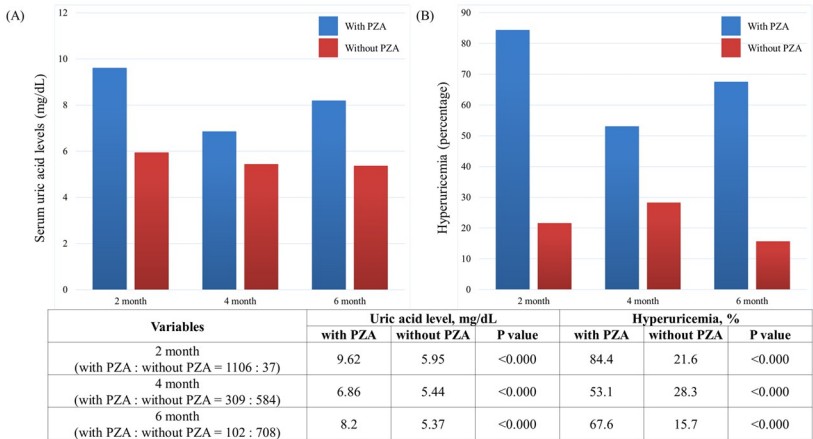

| Variables | Uric acid level, mg/dL | | | Hyperuricemia, % | | |
|---|---|---|---|---|---|---|
| | with PZA | without PZA | P value | with PZA | without PZA | P value |
| 2 month (with PZA : without PZA = 1106 : 37) | 9.62 | 5.95 | <0.000 | 84.4 | 21.6 | <0.000 |
| 4 month (with PZA : without PZA = 309 : 584) | 6.86 | 5.44 | <0.000 | 53.1 | 28.3 | <0.000 |
| 6 month (with PZA : without PZA = 102 : 708) | 8.2 | 5.37 | <0.000 | 67.6 | 15.7 | <0.000 |

**Fig 2. Serum uric acid levels (A) and frequencies of hyperuricemia (B) in patients with or without pyrazinamide (PZA).** The serum uric acid levels (mg/dL) and hyperuricemia incidence (%) in patients with PZA (blue) versus those without PZA (red) at two, four, and six months post-TB treatment initiation are depicted in the accompanying chart, which shows the mean values in bar graph form.

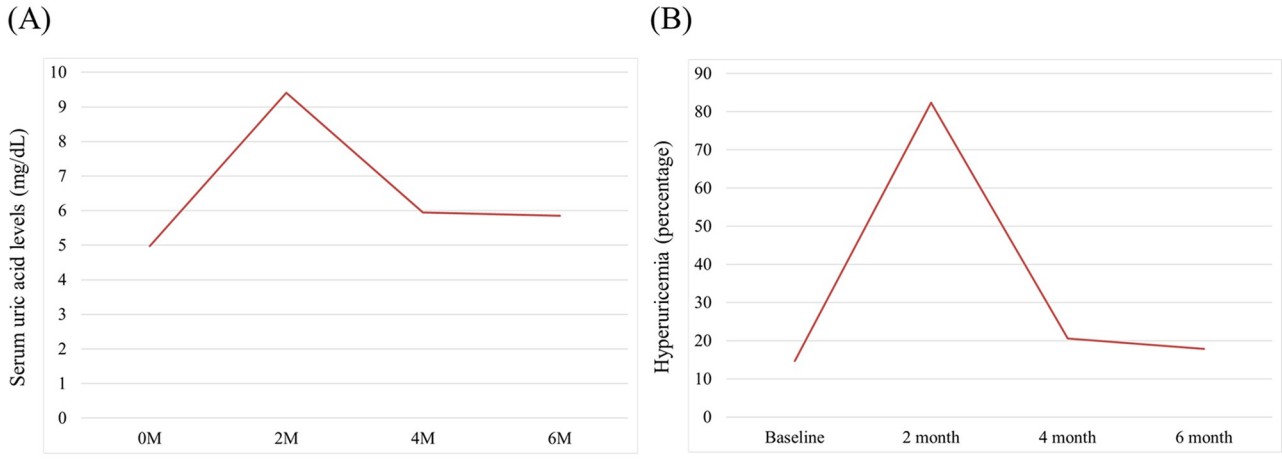

**Fig 3. Serum uric acid levels (A) and patients with hyperuricemia (B) among tuberculous patients whose baseline uric acid levels were measured.**

### Comparisons of baseline characteristics between patients with and without hyperuricemia

Table 1 presents the comparisons between patients with and without hyperuricemia. The mean age and gender distribution were comparable between the two groups. Patients with hyperuricemia more frequently received PZA at the start of TB treatment and exhibited higher hemoglobin levels and lower CRP levels than those without hyperuricemia. However, more patients without hyperuricemia had DM compared to those with hyperuricemia. There was no significant difference in the frequency of drug-resistant TB between the groups.

### The relationship between hyperuricemia and major adverse cardiac events during tuberculosis treatment

Eight patients experienced MACEs during their TB treatment. Both DM and prior ischemic heart disease were significantly more common among patients with MACEs (Table 2). Multivariate analysis using logistic regression revealed that only a history of ischemic heart disease was associated with the development of MACEs (odds ratio [OR]14.087; 95% confidence interval (CI),3.304–60.061; $P < 0.000$) (Table 3). MACEs were not associated with hyperuricemia. However, in female patients, hyperuricemia within 2 months was associated with an increased risk of MACEs (OR, 1.392; 95% CI, 1.021–1.888; $P = 0.036$) (Fig 4).

### Hyperuricemia and survival during tuberculosis treatment

Throughout the study period, 37 patients died while undergoing TB treatment. Multivariate logistic regression analysis revealed old age, low BMI, chronic kidney disease, drug-resistant TB, high CRP levels, and the absence of hyperuricemia as factors associated with death during TB treatment (Table 4). However, hyperuricemia was not associated with death during TB

**Table 1. Baseline characteristics of patients with and without hyperuricemia.**

| Characteristics | With hyperuricemia (n = 941) | Without hyperuricemia (n = 202) | P value |
|---|---|---|---|
| Age, year, mean (SD) | 56.8 (19.4) | 58.4 (17.7) | 0.246 |
| Male, n (%) | 575 (61.1) | 122 (60.4) | 0.874 |
| BMI, mean (SD) | 21.5 (3.6) | 21.1 (3.3) | 0.243 |
| Underlying diseases | | | |
| Diabetes mellitus, n (%) | 137 (14.6) | 62 (30.7) | < 0.000 |
| Hypertension, n (%) | 235 (25.0) | 43 (21.3) | 0.280 |
| Dyslipidemia, n (%) | 52 (5.5) | 10 (5.0) | 0.865 |
| Previous ischemic heart disease, n (%) | 52 (5.5) | 8 (4.0) | 0.486 |
| Atrial fibrillation, n (%) | 26 (2.8) | 4 (2.0) | 0.635 |
| Chronic kidney disease, n (%) | 29 (3.1) | 9 (4.5) | 0.384 |
| Chronic obstructive pulmonary disease, n (%) | 58 (6.2) | 14 (6.9) | 0.635 |
| Gout, n (%) | 22 (2.3) | 3 (1.5) | 0.601 |
| TB classification | | | 0.962 |
| Pulmonary TB, n (%) | 721 (76.5) | 153 (75.7) | |
| Extra-pulmonary TB, n (%) | 165 (17.5) | 37 (18.3) | |
| Pulmonary TB + Extrapulmonary TB, n (%) | 55 (5.8) | 12 (5.9) | |
| Drug resistant TB, n (%) | 85 (9.1) | 20 (10.0) | 0.689 |
| Initial regimen | | | < 0.000 |
| HREZ, n (%) | 911 (96.8) | 169 (83.7) | |
| HRE, n (%) | 7 (0.7) | 20 (9.9) | |
| Other regimens, n (%) | 23 (2.4) | 13 (6.4) | |
| Initial PZA alone based, n (%) | 923 (99.1) | 173 (85.6) | < 0.000 |
| Baseline Hb, mean (SD) | 12.7 (1.7) | 12.2 (1.9) | 0.002 |
| Baseline CRP, mean (SD) | 3.4 (4.5) | 5.0 (6.0) | 0.002 |
| Baseline Cr, mean (SD) | 0.85 (0.66) | 0.94 (1.19) | 0.347 |

Data are presented as the mean (standard deviation) or number (%).

BMI, body mass index; CRP, C-reactive protein; Cr, creatinine; Hb, hemoglobin; HREZ, isoniazid, rifampin, ethambutol, and pyrazinamide; HRE, isoniazid, rifampin, and ethambutol; PZA, pyrazinamide; SD, standard deviation; TB, tuberculosis.

treatment among patients with drug-resistant TB. Among patients without drug-resistant TB, normouricemia was associated with increased mortality during TB treatment (OR, 2.609; 95% CI, 1.066–6.389; P = 0.036).

## Hyperuricemia and treatment outcomes during tuberculosis treatment

Of the 1,052 patients with TB, the majority exhibited favorable treatment outcomes during the study period. Multivariate logistic regression analysis indicated that the presence of chronic kidney disease, drug-resistant TB, baseline anemia, high initial CRP levels, and normouricemia were independently associated with worse outcomes (Table 5).

## Discussion

In our large cohort study, we ascertained the clinical significance of hyperuricemia in relation to MACEs, survival rates, and treatment outcomes in TB patients. The majority of TB patients receiving PZA-containing regimens developed hyperuricemia within the first 2 months of TB treatment. Following the discontinuation of PZA, uric acid levels decreased in most patients. The presence of hyperuricemia during TB treatment was not associated with MACEs.

**Table 2. Comparisons between patients with and without major adverse cardiac events.**

| Characteristics | with MACEs (n = 8) | without MACEs (n = 1,135) | P value |
|---|---|---|---|
| Age, year, mean (SD) | 71.3 (10.5) | 57.0 (19.1) | 0.025 |
| Age, year, median (IQR) | 74.5 (17.7–76.5) | 60.0 (42.0–74.0) | 0.026 |
| Male, n (%) | 5 (62.5) | 693 (61.0) | 1.000 |
| BMI, mean (SD) | 22.3 (4.1) | 21.4 (3.5) | 0.705 |
| BMI, median (IQR) | 21.2 (18.4–26.2) | 21.2 (19.1–23.7) | |
| Underlying diseases | | | |
| Diabetes mellitus, n (%) | 4 (50.0) | 195 (17.2) | 0.035 |
| Hypertension, n (%) | 4 (50.0) | 274 (24.1) | 0.104 |
| Dyslipidemia, n (%) | 2 (25.0) | 60 (5.3) | 0.066 |
| Previous ischemic heart disease, n (%) | 4 (50.0) | 56 (4.9) | < 0.000 |
| Atrial fibrillation, n (%) | 1 (12.5) | 29 (2.6) | 0.192 |
| Chronic kidney disease, n (%) | 1 (12.5) | 37 (3.3) | 0.237 |
| Chronic obstructive pulmonary disease, n (%) | 1 (12.5) | 71 (6.3) | 0.406 |
| Gout, n (%) | 0 (0) | 25 (2.2) | 1.000 |
| TB classification | | | 0.697 |
| Pulmonary TB, n (%) | 7 (87.5) | 868 (76.4) | |
| Extra-pulmonary TB, n (%) | 1 (12.5) | 201 (17.7) | |
| Pulmonary TB + Extrapulmonary TB, n (%) | 0 (0) | 67 (5.9) | |
| Drug resistant TB, n (%) | 2 (25.0) | 103 (9.2) | 0.165 |
| Initial regimen | | | 0.791 |
| HREZ, n (%) | 8 (100) | 1073 (94.5) | |
| HRE, n (%) | 0 (0) | 27 (2.4) | |
| Other regimens, n (%) | 0 (0) | 36 (3.2) | |
| Initial PZA alone based, n (%) | 8 (100) | 1099 (96.7) | 1.000 |
| Baseline Hb, mean (SD) | 12.1 (2.1) | 12.6 (1.8) | 0.932 |
| Baseline Hb, median (IQR) | 11.9 (9.9–13.9) | 12.7 (11.4–13.9) | 0.440 |
| Baseline CRP, mean (SD) | 3.4 (3.7) | 3.7 (4.9) | 0.779 |
| Baseline CRP, median (IQR) | 0.8 (0.4–7.1) | 1.5 (0.4–5.1) | 0.923 |
| Baseline Cr, mean (SD) | 0.90 (0.45) | 0.87 (0.77) | 0.180 |
| Baseline Cr, median (IQR) | 0.70 (0.50–1.10) | 0.70 (0.60–0.90) | 0.776 |
| Hyperuricemia | 7 (87.5) | 934 (82.3) | 1.000 |
| Favorable outcomes | 6 (75.0) | 1040 (94.4) | 0.073 |
| Death | 1 (12.5) | 36 (3.2) | 0.232 |

Data are presented as the median (interquartile range), mean (standard deviation) or number (%).

BMI, body mass index; CRP, C-reactive protein; Cr, creatinine; Hb, hemoglobin; HREZ, isoniazid, rifampin, ethambutol, and pyrazinamide; HRE, isoniazid, rifampin, and ethambutol; MACE, major adverse cardiac events; PZA, pyrazinamide; SD, standard deviation; TB, tuberculosis.

Hyperuricemia during TB treatment was associated with better survival and favorable treatment outcomes.

The emergence of hyperuricemia during TB treatment is primarily attributable to PZA. PZA elevates serum uric acid levels by impeding renal uric acid clearance [3, 4]. A daily dose of 300 mg of PZA leads to an 80% reduction in renal uric acid clearance [3]. It has been documented that between 56% and 86% of patients on PZA develop hyperuricemia [20–22]. Consistent with previous studies, patients who were taking PZA in our study had significantly higher serum uric acid levels within the initial 2 months of treatment (mean, 9.6 mg/dL versus 5.9 mg/dL; P < 0.000) than patients not on PZA. Moreover, 82.3% of this subset developed

**Table 3. Risk factors associated with major adverse cardiac events.**

| Characteristics | OR | 95% CI | P value | OR | 95% CI | P value |
|---|---|---|---|---|---|---|
| | | Univariate analysis | | | Multivariate analysis | |
| Age > 65 years | 9.415 | 1.155–76.778 | 0.036 | | | |
| Male | 1.065 | 0.253–4.480 | 0.931 | | | |
| Body mass index < 18.5 kg/m$^2$ | 1.665 | 0.333–8.320 | 0.535 | | | |
| Diabetes mellitus | 4.826 | 1.197–19.461 | 0.027 | | | |
| Hypertension | 3.146 | 0.782–12.663 | 0.107 | | | |
| Dyslipidemia | 5.978 | 1.181–30.245 | 0.031 | | | |
| Previous ischemic heart disease | 19.286 | 4.701–79.127 | < 0.000 | 14.087 | 3.304–60.061 | < 0.000 |
| Atrial fibrillation | 5.453 | 0.650–45–772 | 0.118 | | | |
| Chronic kidney disease | 4.243 | 0.509–35.377 | 0.182 | | | |
| Chronic obstructive pulmonary disease | 2.143 | 0.260–17.657 | 0.479 | | | |
| Drug resistant TB | 3.304 | 0.658–16.581 | 0.146 | | | |
| Hyperuricemia | 1.505 | 0.184–12.299 | 0.703 | | | |

CI, confidence interval; MACE, major adverse cardiac event; OR, odds ratio; TB, tuberculosis.

hyperuricemia. It is worth noting that PZA-induced hyperuricemia is transient. The serum uric acid levels are known to decline after the discontinuation of PZA [22]. Our research corroborated this, as we observed that serum uric acid levels declined after PZA discontinuation among individuals who were on PZA and had hyperuricemia. Therefore, hyperuricemia, in

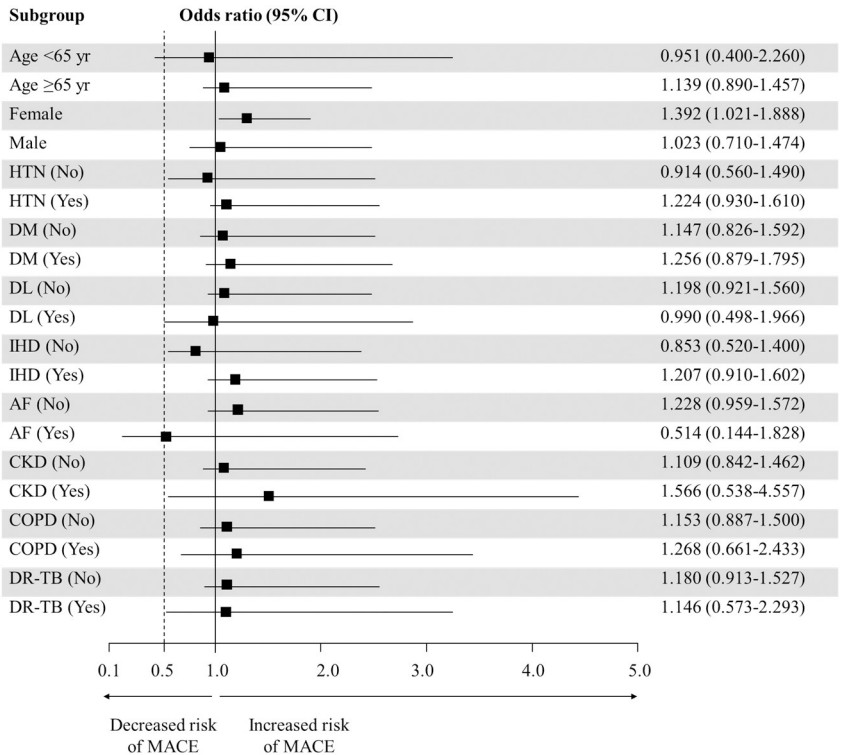

**Fig 4. Subgroup analysis of hyperuricemia.** AF, atrial fibrillation; COPD, chronic obstructive pulmonary disease; CKD, chronic kidney disease; DR-TB, drug resistant tuberculosis; IHD, ischemic heart disease; MACE, major adverse cardiac event.

**Table 4. Risk factors associated with death during tuberculosis treatment.**

| Characteristics | OR | 95% CI | P value | OR | 95% CI | P value |
|---|---|---|---|---|---|---|
| | | Univariate analysis | | | Multivariate analysis | |
| Age > 65 years | 2.861 | 1.423–5.754 | 0.003 | 2.960 | 1.167–7.510 | 0.022 |
| Male | 2.032 | 0.950–4.348 | 0.068 | | | |
| Body mass index < 18.5 kg/m$^2$ | 2.580 | 1.185–5.618 | 0.017 | 4.395 | 1.734–11.143 | 0.002 |
| Diabetes mellitus | 1.550 | 0.720–3.337 | 0.263 | | | |
| Hypertension | 1.516 | 0.751–3.059 | 0.246 | | | |
| Dyslipidemia | 1.566 | 0.467–5.247 | 0.467 | | | |
| Previous ischemic heart disease | 1.003 | 0.242–4.399 | 0.965 | | | |
| Atrial fibrillation | 3.526 | 1.020–12.193 | 0.047 | | | |
| Chronic kidney disease | 6.496 | 2.532–16.667 | < 0.000 | 7.146 | 1.980–25.794 | 0.003 |
| Chronic obstructive pulmonary disease | 1.850 | 0.637–5.374 | 0.258 | | | |
| Drug resistant TB | 2.371 | 1.015–5.540 | 0.046 | 4.141 | 1.309–13.099 | 0.016 |
| Initial PZA alone based | 0.359 | 0.105–1.228 | 0.103 | | | |
| Baseline anemia | 3.891 | 1.723–8.791 | 0.001 | | | |
| Baseline C-reactive protein | 1.123 | 1.072–1.177 | < 0.000 | 1.125 | 1.060–1.193 | < 0.000 |
| Baseline Creatinine | 1.384 | 1.118–1.714 | 0.003 | | | |
| No hyperuricemia | 3.769 | 1.930–7.359 | < 0.000 | 2.609 | 1.066–6.389 | 0.036 |

CI, confidence interval; OR, odds ratio; PZA, pyrazinamide; TB, tuberculosis.

the context of standard TB treatment with 2 months of PZA administration, is short-lived and quickly resolves following the discontinuation of PZA.

Hyperuricemia is a recognized risk factor for cardiovascular diseases [8–10]. Elevated serum uric acid levels are associated with an increased incidence of MACEs and higher

**Table 5. Risk factors associated with worse outcomes in patients with tuberculosis treatment.**

| Characteristics | OR | 95% CI | P value | OR | 95% CI | P value |
|---|---|---|---|---|---|---|
| | | Univariate analysis | | | Multivariate analysis | |
| Age > 65 years | 1.610 | 1.047–2.475 | 0.030 | | | |
| Male | 1.026 | 0.661–1.593 | 0.909 | | | |
| Body mass index < 18.5 kg/m$^2$ | 1.736 | 0.990–3.045 | 0.054 | | | |
| Diabetes mellitus | 1.795 | 1.096–2.941 | 0.020 | | | |
| Hypertension | 1.196 | 0.739–1.936 | 0.466 | | | |
| Dyslipidemia | 1.015 | 0.396–2.599 | 0.975 | | | |
| Previous ischemic heart disease | 0.595 | 0.183–1.939 | 0.389 | | | |
| Atrial fibrillation | 3.024 | 1.203–7.599 | 0.019 | | | |
| Chronic kidney disease | 3.872 | 1.773–8.454 | 0.001 | 4.079 | 1.581–10.527 | 0.004 |
| Chronic obstructive pulmonary disease | 1.265 | 0.562–2.847 | 0.569 | | | |
| Drug resistant TB | 2.783 | 1.587–4.881 | < 0.000 | 3.585 | 1.612–7.975 | 0.002 |
| Initial PZA alone based | 0.213 | 0.100–0.456 | < 0.000 | | | |
| Baseline anemia | 3.351 | 2.006–5.599 | < 0.000 | 2.285 | 1.221–4.278 | 0.010 |
| Baseline C-reactive protein | 1.076 | 1.037–1.117 | < 0.000 | 1.054 | 1.006–1.104 | 0.026 |
| Baseline Creatinine | 1.223 | 1.002–1.493 | 0.048 | | | |
| Hyperuricemia | 0.319 | 0.202–0.502 | < 0.000 | 0.316 | 0.173–0.576 | < 0.000 |

CI, confidence interval; OR, odds ratio; PZA, pyrazinamide; TB, tuberculosis.

mortality rates in coronary heart diseases [23–25]. Elevated serum uric acid levels might diminish the effectiveness of medications like losartan, fenofibrate, and atorvastatin [26]. While the precise mechanism linking elevated serum uric acid levels to an increased risk of cardiovascular disease remains undetermined, several plausible explanations exist. High serum uric acid levels may increase blood pressure by stimulating the renin-angiotensin system, diminishing endothelial nitric oxide secretion, and obstructing renal vasoconstrictions [8]. Such levels can also contribute to metabolic syndrome [8, 27, 28] and chronic kidney disease [8]. Notably, in this study, hyperuricemia was not associated with MACEs. It is possible that only chronic hyperuricemia elevates the risk of cardiovascular disease [28]. Most hyperuricemia during TB treatment in this study may not have lasted long enough to lead to MACEs, especially since serum uric acid levels declined after the 2-month PZA treatment. As such, there may be no need to discontinue PZA or administer serum uric acid-reducing agents for patients with hyperuricemia during standard TB treatment, as transient hyperuricemia may not be associated with MACEs. However, in TB patients undergoing long-term PZA treatment, further investigation is needed to determine if hyperuricemia persisting beyond 2 months is associated with MACEs.

Cardiovascular disease is more common in men than in women [29]. In contrast, the associations between elevated uric acid levels and MACEs have been shown to be higher in women than men [30]. Even though a definitive explanation is lacking, sex hormone and lifestyle differences have been suggested to explain this difference [16]. Consistent with previous research, our study showed that women had a significantly higher risk of MACEs in the subgroup analysis of patients with hyperuricemia. Further research is warranted to determine whether MACEs occur more frequently in females with hyperuricemia during TB treatment.

Adherence to treatment is crucial for controlling TB, as low adherence can escalate the risk of treatment failure, relapse, and the emergence of drug resistance. However, adherence is multifaceted, and several factors can influence it [31, 32]. Consequently, assessing adherence to anti-TB drugs is paramount. However, there are few methods of monitoring adherence to these medications. The TB Case Management Strategy, which includes directly observed treatment (DOT), received endorsement from the WHO in the 1990s [33]. Nonetheless, DOT has not shown superior efficacy in addressing poor adherence when compared with self-administered therapy [34]. Lately, a variety of digital tools designed to improve self-administered therapy using phone- or smartphone-based digital methods have emerged, with research into their efficacy still ongoing [35]. In this study, hyperuricemia during TB treatment was associated with better outcomes and survival. Given that hyperuricemia can arise from PZA intake, the presence of hyperuricemia during TB treatment might signify good adherence to anti-TB drugs, potentially offering a straightforward adherence monitoring tool. During TB treatment, hyperuricemia was not associated with MACEs. Hyperuricemia is hypothesized to be associated with good adherence to anti-TB medications. Notably, normouricemia was associated with increased mortality during TB treatment among patients without drug-resistant TB.

Several limitations were present in this study. First, this study was conducted retrospectively at a single center; therefore, the generalization of our findings could be limited. Second, only 32.7% of all patients had baseline uric acid levels; therefore, baseline hyperuricemia could not be confirmed in 77.3% of patients, leaving us uninformed regarding their pre-treatment status. Therefore, we could not determine the proportion of patients with preexisting hyperuricemia prior to the start of anti-TB medication. Third, hyperuricemia might be influenced by dietary intake, including fructose or purine-rich meats, and by certain drugs that curtail uric acid clearance. The impact of such foods or medications was not evaluated in this study. Therefore, it is unclear whether hyperuricemia persisting after PZA discontinuation is a continuation of baseline hyperuricemia or an increase in uric acid production due to meat consumption.

## Conclusion

Most patients who received PZA-based anti-TB drugs had hyperuricemia; however, hyperuricemia was not associated with the development of MACEs. Hyperuricemia during the treatment for TB was associated with better outcomes, possibly due to good compliance with anti-TB drugs.

## Supporting information

**S1 Dataset.**
(XLSX)

**S1 Fig.**
(TIF)

**S2 Fig.**
(TIF)

## Author Contributions

**Conceptualization:** Hong-Joon Shin, Yong-Soo Kwon.

**Data curation:** Hong-Joon Shin, Joon-Young Yoon, Young-Ok Na, Jae-Kyeong Lee, Bo Gun Kho, Tae-Ok Kim, Sae-Hee Jeong.

**Formal analysis:** Hong-Joon Shin, Yu-Il Kim, Sung-Chul Lim, Yong-Soo Kwon.

**Funding acquisition:** Yong-Soo Kwon.

**Methodology:** Hong-Joon Shin.

**Writing – original draft:** Hong-Joon Shin, Yong-Soo Kwon.

**Writing – review & editing:** Hong-Joon Shin, Joon-Young Yoon, Young-Ok Na, Jae-Kyeong Lee, Bo Gun Kho, Tae-Ok Kim, Yu-Il Kim, Sung-Chul Lim, Sae-Hee Jeong, Yong-Soo Kwon.

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
