## [Decision Letter · Decision Letter 0]

27 Sep 2023

PONE-D-23-24904Major adverse cardiovascular events and hyperuricemia during tuberculosis treatmentPLOS ONE

Dear Dr. Kwon,

Thank you for submitting your manuscript to PLOS ONE. After careful consideration, we feel that it has merit but does not fully meet PLOS ONE’s publication criteria as it currently stands. Therefore, we invite you to submit a revised version of the manuscript that addresses the points raised during the review process.

Please submit your revised manuscript by Nov 11 2023 11:59PM. If you will need significantly more time to complete your revisions, please reply to this message or contact the journal office at plosone@plos.org. Please include the following items when submitting your revised manuscript:A rebuttal letter that responds to each point raised by the academic editor and reviewer(s). You should upload this letter as a separate file labeled 'Response to Reviewers'.A marked-up copy of your manuscript that highlights changes made to the original version. You should upload this as a separate file labeled 'Revised Manuscript with Track Changes'.An unmarked version of your revised paper without tracked changes. You should upload this as a separate file labeled 'Manuscript'.

We look forward to receiving your revised manuscript.

Kind regards,

Frederick Quinn

Academic Editor

PLOS ONE

[This study was supported by a grant (BCRI20013) of Chonnam National University Hospital Biomedical Research Institute. The funders had no role in study design, data collection and analysis, decision to publish, or preparation of the manuscript.]

 [YSK received a grant (BCRI20013) of Chonnam National University Hospital Biomedical Research Institute. The funders had no role in study design, data collection and analysis, decision to publish, or preparation of the manuscript]

5. Please remove your figures from within your manuscript file, leaving only the individual TIFF/EPS image files, uploaded separately. These will be automatically included in the reviewers’ PDF.

Reviewers' comments:

Reviewer's Responses to Questions

**Comments to the Author**

1. Is the manuscript technically sound, and do the data support the conclusions?

Reviewer #1: Yes

2. Has the statistical analysis been performed appropriately and rigorously? 

Reviewer #1: Yes

3. Have the authors made all data underlying the findings in their manuscript fully available?

Reviewer #1: Yes

4. Is the manuscript presented in an intelligible fashion and written in standard English?

Reviewer #1: Yes

5. Review Comments to the Author

Reviewer #1: 1. Whole manuscript should be reviewed by a native speaker.

2. The aim of the study should be added into the last paragraph of introduction section.

3.The authors stated that ETM causes hyperuricemia in the introduction section. It should be checked.

4. The authors stated that ETM could be given during 6 months in their treatment protocol. If ETM causes hyperuricemia, patients who receive ETM for 9 months and who did not should be compared in terms of all results.

5. How did the authors explain the hyperuricemia in the fourth and sixth months of the treatment?

6. It would be better if the authors add some comments why female patients had higher risk for MACEs.

7. Why does the absence of hyperuricemia increase the risk of death? The authors may comment on this issue in the discussion section.

8. Title of Table 5 is confusing. It would be better to give risk factors for worse outcomes instead of risk factors for favorable outcomes. The authors should calculate risk factors for worse aoutcomes and they should be displayed in the table.

6. PLOS authors have the option to publish the peer review history of their article (what does this mean?). If published, this will include your full peer review and any attached files.

Reviewer #1: No

---

## [Author Response · Author response to Decision Letter 0]

30 Oct 2023

PONE-D-23-24904

Major adverse cardiovascular events and hyperuricemia during tuberculosis treatment

Frederick Quinn

Academic Editor

PLOS ONE

Dear Dr. Quinn, 

Thank you for your letter dated August 28, 2023. We appreciate you and the reviewers for reviewing our manuscript entitled “Article Title: Major adverse cardiovascular events and hyperuricemia during tuberculosis treatment” and offering helpful suggestions. We are submitting a revised manuscript that addresses the concerns that the reviewers raised. A detailed point-by-point response follows below. 

We look forward to any additional comments or questions concerning this paper and hope that the revised manuscript is now acceptable for publication in PLOS ONE.

Sincerely,

Yong-Soo Kwon, M.D., Ph.D., on behalf of all the authors.

Department of Internal Medicine

Chonnam National University Hospital

Gwangju, Korea

E-mail: yskwon@jnu.ac.kr

Editor, PLOS ONE

Reviewer #1: 

Q1. Whole manuscript should be reviewed by a native speaker.

A1. The manuscript was edited by an English editing service. 

Q2. The aim of the study should be added into the last paragraph of introduction section.

A2. Thank you for your valuable comment. This study's purpose was described in the introduction. “This study aimed to evaluate the association between major adverse cardiovascular events (MACEs) and hyperuricemia during TB treatment.”

Q3. The authors stated that ETM causes hyperuricemia in the introduction section. It should be checked.

A3. According to the literature, EMB also reduces renal uric acid excretion and raises serum uric acid levels. However, ethambutol is infrequently associated with hyperuricemia, and its effect on serum uric acid is minimal compared with pyrazinamide. The reference and the mentioned content have been added to the introduction.

Q4. The authors stated that ETM could be given during 6 months in their treatment protocol. If ETM causes hyperuricemia, patients who receive ETM for 9 months and who did not should be compared in terms of all results.

A4. Thank you for your valuable comments. First of all, the frequencies of hyperuricemia significantly varied according to the regimen (see Table 1); the frequency of hyperuricemia associated with a regimen with EMB and without PZA (HRE) was only 25.9% (7/27) which was significantly lower than that associated with HREZ (84.3% [911/1080]). Therefore, the association with EMB and hyperuricemia could be much lower than that associated with PZA and hyperuricemia.

Additionally, we performed a statistical analysis to determine whether the EMB could be associated with hyperuricemia. Among patients receiving either HRE or HR treatment after 2 months of TB treatment, serum uric acid levels and hyperuricemia were assessed at 4 and 6 months. In Figures S1 and S2, serum uric acid levels and hyperuricemia did not differ significantly between the HRE and HR groups. Accordingly, ethambutol did not significantly affect serum uric acid levels in this study. The mentioned content has been added to the results section.

Q5. How did the authors explain the hyperuricemia in the fourth and sixth months of the treatment?

A5. Figure 3 shows the changes in serum uric acid levels during TB treatment. In this figure, uric acid levels and the frequency of hyperuricemia peaked in the second month and then decreased sharply, almost reaching baseline in the fourth and sixth months of TB treatment. Therefore, the hyperuricemia in the fourth and sixth months of TB treatment might have been caused by factors other than PZA.

Q6. It would be better if the authors add some comments why female patients had higher risk for MACEs.

A6. Cardiovascular disease is more common in men than in women [Atherosclerosis . 2015 Jul;241(1):211-8. #25670232]. In contrast, the associations between elevated uric acid levels and MACE have been shown to be higher in women than men. [J Cardiol . 2016 Feb;67(2):170-6. #26228000]. Even though a definitive explanation is lacking, sex hormone and lifestyle differences have been suggested to explain this difference. [ Lipids Health Dis . 2018 Apr 11;17(1):80. #29642917]. Further research is warranted. The mentioned content has been added to the Discussion section.

Q7. Why does the absence of hyperuricemia increase the risk of death? The authors may comment on this issue in the discussion section.

A7. Hyperuricemia could imply a good adherence to anti-TB drugs since it indirectly indicates high adherence to PZA. Therefore, the absence of hyperuricemia could be associated with poor adherence to anti-TB drugs (and, therefore, with death). We have included this explanation in the Discussion section.

“In this study, hyperuricemia during TB treatment was associated with better outcomes and survival. Given that hyperuricemia can arise from PZA intake, the presence of hyperuricemia during TB treatment might signify good adherence to anti-TB drugs, potentially offering a straightforward adherence monitoring tool.”

Q8. Title of Table 5 is confusing. It would be better to give risk factors for worse outcomes instead of risk factors for favorable outcomes. The authors should calculate risk factors for worse outcomes and they should be displayed in the table.

A8. Thank you for your comments. We have modified Table 5 according to your recommendations.

---

## [Decision Letter · Decision Letter 1]

2 Nov 2023

Major adverse cardiovascular events and hyperuricemia during tuberculosis treatment

PONE-D-23-24904R1

Dear Dr. Kwon,

We’re pleased to inform you that your manuscript has been judged scientifically suitable for publication and will be formally accepted for publication once it meets all outstanding technical requirements.

Kind regards,

Frederick Quinn

Academic Editor

PLOS ONE

Additional Editor Comments (optional):

Reviewers' comments:

Reviewer's Responses to Questions

**Comments to the Author**

1. If the authors have adequately addressed your comments raised in a previous round of review and you feel that this manuscript is now acceptable for publication, you may indicate that here to bypass the “Comments to the Author” section, enter your conflict of interest statement in the “Confidential to Editor” section, and submit your "Accept" recommendation.

Reviewer #1: All comments have been addressed

2. Is the manuscript technically sound, and do the data support the conclusions?

Reviewer #1: Yes

3. Has the statistical analysis been performed appropriately and rigorously? 

Reviewer #1: Yes

4. Have the authors made all data underlying the findings in their manuscript fully available?

Reviewer #1: Yes

5. Is the manuscript presented in an intelligible fashion and written in standard English?

Reviewer #1: Yes

6. Review Comments to the Author

Reviewer #1: (No Response)

7. PLOS authors have the option to publish the peer review history of their article (what does this mean?). If published, this will include your full peer review and any attached files.

Reviewer #1: No

---

## [Editor Report · Acceptance letter]

8 Nov 2023

PONE-D-23-24904R1 

Major adverse cardiovascular events and hyperuricemia during tuberculosis treatment 

Dear Dr. Kwon:

I'm pleased to inform you that your manuscript has been deemed suitable for publication in PLOS ONE. Congratulations! Your manuscript is now with our production department. 

Kind regards, 

on behalf of

Dr. Frederick Quinn 

Academic Editor

PLOS ONE